# Contrastive VQ Priors for Multi-Class Plaque Segmentation via SAM Adaptation

**Yizhe Ruan**[1,2]**, Yusuke Kurose**[1,2]**, Junichi Iho**[3]**,**
**Yoji Tokunaga**[3]**, Makoto Horie**[3]**, Yusaku Hayashi**[3]**, Keisuke Nishizawa**[3]**,**
**Yasushi Koyama**[3,2]**, Tatsuya Harada**[1,2]
[1]**The University of Tokyo**
[2]**RIKEN Center for Advanced Intelligence Project**
[3]**Sakurabashi Watanabe Advanced Healthcare Hospital**
ruanyizhe@mi.t.u-tokyo.ac.jp

Reviewed on OpenReview: https://openreview.net/forum?id=5P7HfuejgL

## Abstract

Accurate plaque subtype segmentation in coronary CT angiography (CCTA) is clinically relevant yet remains difficult in practice, where annotations are scarce and the visual evidence for non-calcified lesions is subtle and highly variable. Meanwhile, segmentation foundation models such as SAM provide strong robustness from large-scale pretraining, but their benefits do not reliably transfer to private CCTA tasks under naïve fine-tuning, especially for multi-class plaque taxonomy. We present a targeted strategy to transfer SAM's segmentation robustness to a private CCTA setting by injecting a task-specific, texture-aware prior into the SAM feature stream. Our framework is two-stage: (i) we learn a discrete latent prior from the private CCTA data using a vector-quantized autoencoder, and structure it with supervised contrastive learning to emphasize hard class boundaries; (ii) we fuse this prior into a SAM-based encoder through a query-based feature-aware cross-attention module, and decode with a multi-class head/decoder tailored for plaque taxonomy. On this private CCTA cohort, the proposed design improves overall performance over the compared baselines, with the largest gains on vessel wall and non-calcified plaque. Ablations suggest that the class-structured prior, query-based fusion, and multi-class decoding each contribute to the final result within this setting.

## 1 Introduction

Coronary CT angiography (CCTA) enables non-invasive assessment of coronary artery disease and is widely used for characterizing atherosclerotic plaques and guiding risk stratification (Serruys et al., 2021). A key step toward quantitative analysis is segmenting plaque regions and distinguishing clinically meaningful subtypes (e.g., calcified vs. non-calcified). However, plaque segmentation in CCTA is a stubborn problem: lesions can be small, boundaries are often weak due to partial-volume effects and imaging artifacts, and intra-class appearance varies substantially across patients and acquisition conditions. In private clinical settings, these challenges are compounded by limited expert annotations and restricted data sharing, making it difficult to rely solely on large supervised training sets.

Classic medical segmentation pipelines remain strong baselines when sufficient labeled data are available. Self-configuring systems such as nnU-Net reduce manual design choices and often deliver competitive performance across tasks (Isensee et al., 2021), while transformer-based hybrids such as TransUNet improve global context modeling (Chen et al., 2021). Yet for plaque subtypes in CCTA, the discriminative evidence is frequently texture-level and sparse, so standard supervised training can fall into class confusion (e.g., wall vs. plaque) or overlook subtle non-calcified regions even when the global anatomy is captured well.

Recent segmentation foundation models offer a complementary direction. SAM demonstrates remarkable robustness and generalization from large-scale pretraining (Kirillov et al., 2023). Medical variants adapt SAM to clinical imagery via domain-specific training recipes and weights (Ma et al., 2024; Zhang & Liu, 2023), and parameter-efficient tuning strategies have been developed to stabilize adaptation under limited supervision (Xiao et al., 2024). Nevertheless, transferring foundation-model robustness to a *private* CCTA multi-class plaque task is not automatic. Naïve fine-tuning can overfit to the limited labeled distribution, while frozen-feature approaches may not expose sufficient task-specific cues for plaque subtype separation. In our experiments, we specifically base our approach on *SAM* rather than a purely medical-specialized initialization, motivated by the practical observation that SAM's broad pretraining tends to retain stronger transferability when adapting to specialized sub-tasks with domain (Li & Rajpurkar, 2024; Chao et al., 2025); our goal is therefore not to rebuild a CCTA model from scratch, but to *selectively pull* the right CCTA-specific evidence into a robust pretrained segmentation backbone.

We propose a targeted transfer strategy that augments a SAM-based encoder with a *texture-aware prior* learned directly from the private CCTA data. Our key idea is to separate "robust segmentation competence" (carried by SAM) from "task-specific plaque evidence" (learned from the private CCTA distribution), and then connect them through a retrieval-like fusion mechanism. Concretely, we first learn a discrete latent representation using a vector-quantized autoencoder (van den Oord et al., 2017; Razavi et al., 2019; Esser et al., 2021) so that recurring texture patterns relevant to plaque subtypes can be expressed as compact codes. To sharpen the separation between hard classes, we impose supervised contrastive structure on the latent space (Khosla et al., 2020). In the second stage, we inject these discrete priors into the SAM feature stream through a feature-aware cross-attention (FAA) module, where SAM features query the learned priors and retrieve complementary cues only when needed. Finally, we use a multi-class decoding strategy compatible with SAM-style architectures to produce plaque subtype masks.

**Contributions.**

- We introduce a two-stage adaptation pipeline for private multi-class CCTA plaque segmentation, where a class-structured discrete prior learned on the training split is injected into a SAM-based encoder.

- We study a query-based feature-aware cross-attention fusion (FAA) that allows the encoder to retrieve prior features selectively, with the strongest benefit appearing on the hardest plaque categories.

- We compare two practical multi-class output designs for SAM-style segmentation and evaluate the full pipeline against strong non-SAM and SAM-family baselines, together with ablations of the major design choices.

## 2 Related Work

### 2.1 Coronary Plaque Segmentation in CCTA

CCTA plaque segmentation has attracted sustained interest because precise plaque burden and subtype quantification are tightly linked to downstream clinical interpretation (Serruys et al., 2021; Föllmer et al., 2024). Recent systems increasingly combine strong anatomical modeling with lesion-focused cues, but performance remains sensitive to private data distributions and annotation constraints. Specialized pipelines and plaque-oriented models have been explored to improve subtype delineation and robustness in coronary imaging (Wang et al., 2024). Our work shares the same objective—accurate plaque subtype segmentation under realistic constraints—but focuses on leveraging foundation-model robustness while learning a CCTA-specific prior from private data.

### 2.2 Medical Segmentation Baselines

U-Net-style architectures remain a cornerstone of medical segmentation, and nnU-Net in particular offers a strong, self-adapting baseline across diverse datasets (Isensee et al., 2021). Transformer-based variants

such as TransUNet incorporate global context and have shown improved performance on multi-organ and challenging boundary tasks (Chen et al., 2021). We include both as primary baselines because they represent mature, high-performing supervised pipelines that are often deployed when sufficient labels exist, and they provide an important reference for judging whether foundation-model transfer yields tangible gains.

### 2.3 Segmentation Foundation Models and SAM Adaptation

SAM introduced promptable segmentation with large-scale pretraining and strong robustness across visual domains (Kirillov et al., 2023). To better match clinical imagery, Medical SAM variants adapt SAM through medical data and task-specific training recipes (Ma et al., 2024; Zhang & Liu, 2023). In parallel, parameter-efficient adaptation methods aim to transfer SAM with limited supervision while controlling overfitting and preserving generalization; CAT-SAM is a representative approach that conditions SAM with lightweight tuning mechanisms (Xiao et al., 2024). Different from works that primarily optimize adaptation efficiency or interactive prompting behavior, we target *segmentation performance* on a private CCTA multi-class task and ask: how can one retain SAM's robustness while reliably injecting the missing, texture-level evidence needed for plaque subtype separation?

### 2.4 Discrete Latent Priors and Contrastive Structuring

Discrete latent models such as VQ-VAE learn codebooks that represent recurring patterns as compact tokens (van den Oord et al., 2017; Razavi et al., 2019). Later developments (e.g., VQGAN) further improved perceptual fidelity and representation usefulness for downstream tasks (Esser et al., 2021). Separately, supervised contrastive learning provides a powerful mechanism to enforce intra-class compactness and inter-class separation in representation space (Khosla et al., 2020). We build on these ideas to learn a CCTA-specific discrete prior and explicitly shape it so that hard boundaries (notably wall vs. plaque subtypes) become more separable, enabling an attention-based retrieval module to inject the right cues into a robust SAM encoder.

## 3 Method

### 3.1 Problem Setup

Let $x \in \mathbb{R}^{H \times W}$ be a single-channel CCTA slice (after standard HU normalization), and $y \in \{0, \dots, C-1\}^{H \times W}$ the corresponding multi-class label map with $C = 5$ classes: background, lumen, vessel wall, calcified plaque, and non-calcified plaque. Our goal is to learn a segmentation model $f_\theta$ that predicts per-pixel class probabilities $\hat{p} \in [0, 1]^{C \times H \times W}$ with high accuracy, emphasizing plaque subtypes.

### 3.2 Overview

As shown in Figure 1, our approach consists of two stages:

1. **Stage 1: Contrastive discrete latent prior learning.** We train a VQ autoencoder on the private CCTA training set to learn a discrete latent map. In addition to reconstruction and VQ commitment terms, we apply supervised contrastive learning on projected quantized features sampled from hard classes (vessel wall and plaques) to explicitly structure the latent space for wall/plaque discrimination.

2. **Stage 2: Foundation encoder adaptation with Feature-Aware Attention (FAA).** We freeze the VQ encoder and treat its quantized latent map as a domain prior. We adapt a SAM-based encoder for semantic segmentation and fuse SAM embeddings (queries) with VQ priors (keys/values) using a lightweight cross-attention FAA block.

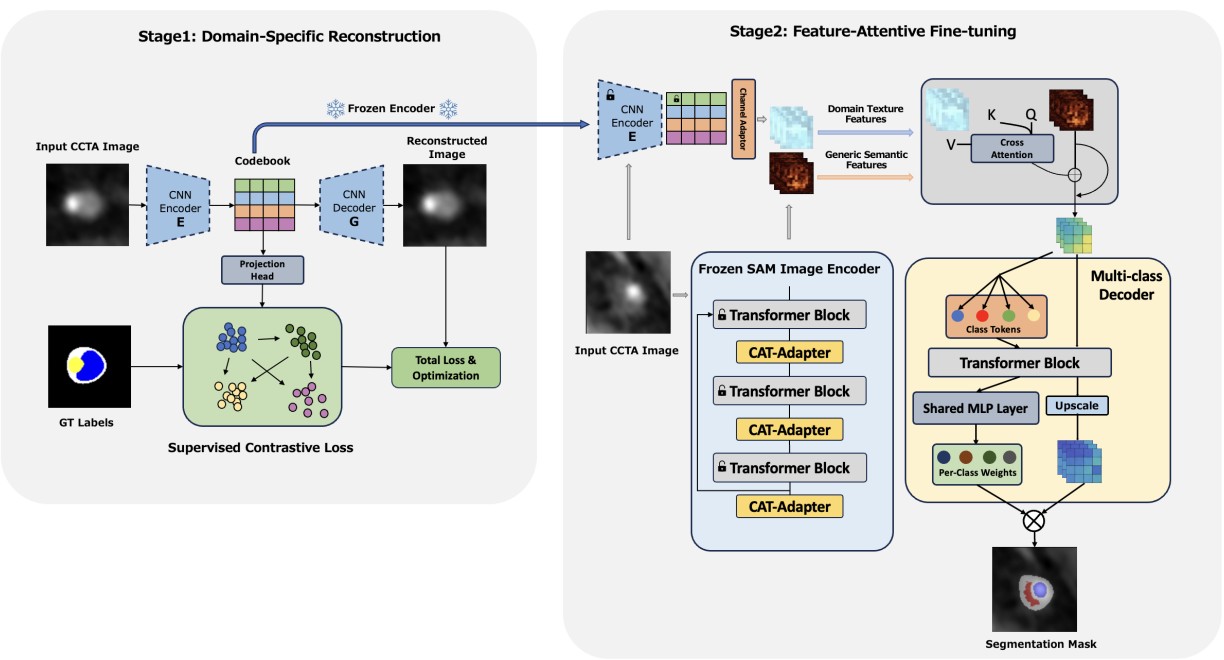

Figure 1: Framework overview. Stage 1 learns a class-structured VQ latent prior with reconstruction + commitment + supervised contrastive objectives. Stage 2 injects frozen VQ priors into a SAM-based encoder via FAA cross-attention for high-performance multi-class plaque segmentation.

### 3.3 Stage 1: Contrastive VQ Autoencoder for Latent Priors

**Discrete latent learning.** We adopt a VQ-VAE architecture to learn a discrete latent representation. Given input slice $x$, an encoder $E_\phi$ produces a continuous feature map

$$h = E_\phi(x) \in \mathbb{R}^{d \times h \times w}, \tag{1}$$

where $(h, w)$ is the downsampled spatial resolution. A codebook $\mathcal{Z} = \{e_k\}_{k=1}^K$ with $K$ embeddings quantizes $h$ into a discrete latent map $z_{\mathrm{vq}}$ by nearest-neighbor assignment:

$$z_{\mathrm{vq}}(i,j) = e_{\arg\min_k \|h(i,j) - e_k\|_2}. \tag{2}$$

A decoder $D_\phi$ reconstructs $\hat{x} = D_\phi(z_{\mathrm{vq}})$.

**Losses: reconstruction, perceptual, commitment.** We optimize

$$\mathcal{L}_{\mathrm{VQ}} = \lambda_1 \|x - \hat{x}\|_1 + \lambda_p \mathcal{L}_{\mathrm{perc}}(x, \hat{x}) + \lambda_c \mathcal{L}_{\mathrm{commit}} + \lambda_{\mathrm{con}} \mathcal{L}_{\mathrm{supcon}}. \tag{3}$$

The commitment loss follows standard VQ training, encouraging stable codebook usage. The perceptual loss $\mathcal{L}_{\mathrm{perc}}$ ( LPIPS) improves texture fidelity and reduces overly smooth reconstructions (Zhang et al., 2018; Johnson et al., 2016), which is important for subtle plaque cues.

**Supervised contrastive learning on projected quantized features.** Vanilla VQ training does not ensure that wall/plaque textures form separable clusters. To enforce class structure, we attach a projection head $P_\phi$ to quantized features and apply supervised contrastive learning. Let

$$u = P_\phi(z_{\mathrm{vq}}) \in \mathbb{R}^{d' \times h \times w} \tag{4}$$

denote projected features. We downsample the ground-truth mask $y$ to latent resolution $\tilde{y} \in \{0, \ldots, C-1\}^{h \times w}$. We then sample $N$ feature vectors from the hard classes

$$\mathcal{C}_{\mathrm{hard}} = \{\mathrm{wall}, \mathrm{calcified}, \mathrm{non\text{-}calcified}\}, \tag{5}$$

forming a set $\mathcal{S} = \{(u_i, \tilde{y}_i)\}_{i=1}^N$. The supervised contrastive objective (InfoNCE-style) is

$$\mathcal{L}_{\text{supcon}} = -\frac{1}{|\mathcal{I}|} \sum_{i \in \mathcal{I}} \log \frac{\sum_{j \in \mathcal{P}(i)} \exp(\text{sim}(u_i, u_j)/\tau)}{\sum_{k \in \mathcal{K}(i)} \exp(\text{sim}(u_i, u_k)/\tau)}, \tag{6}$$

where $\mathcal{P}(i)$ are indices with the same label as $i$ (positives), $\mathcal{K}(i)$ are all indices excluding $i$ (positives+negatives), $\text{sim}(\cdot, \cdot)$ is cosine similarity, and $\tau$ is temperature. Sampling only hard classes prevents the objective from being dominated by easy background/lumen patterns and concentrates discriminative pressure on wall/plaque ambiguity.

**Output as a frozen prior.** After Stage 1 training, we freeze $E_\phi$ and the codebook $\mathcal{Z}$. For any input $x$, we compute $z_{\text{vq}}$ and treat it as a domain prior for Stage 2.

### 3.4 Stage 2: SAM Adaptation with Feature-Aware Attention Fusion

Our goal is not to propose a universally general adaptation rule for SAM across medical domains. Rather, we study whether, for this private CCTA plaque-segmentation setting, a frozen domain prior learned from the training split can complement a strong pretrained encoder through selective feature retrieval.

**SAM-based backbone for semantic segmentation.** We use a SAM image encoder as a strong foundation backbone. Since SAM was originally designed for promptable binary masks, we adapt it to multi-class semantic segmentation by attaching a lightweight segmentation head (or adapting the decoder to multi-class outputs). We also consider stable conditional tuning modules as a baseline adaptation mechanism; however, our focus is not few-shot tuning but improving performance by injecting the learned prior.

Let $F_{\text{sam}} \in \mathbb{R}^{d_s \times H' \times W'}$ denote the SAM embedding map at a chosen resolution (e.g., $H' = W' = 64$ for an embedding grid), with channel dimension $d_s$.

**Channel alignment for VQ priors.** The quantized latent map $z_{\text{vq}} \in \mathbb{R}^{d \times h \times w}$ is resized to $(H', W')$ and mapped to $d_s$ channels using a $1 \times 1$ convolution (channel adapter) $A$:

$$F_{\text{vq}} = A(\text{resize}(z_{\text{vq}})) \in \mathbb{R}^{d_s \times H' \times W'}. \tag{7}$$

**Feature-Aware Attention (FAA): query-based fusion.** A naive additive fusion $F_{\text{sam}} + F_{\text{vq}}$ injects priors everywhere and may harm regions that are already easy. Instead, as shown in Figure 2, we propose FAA, a cross-attention block where SAM embeddings query VQ priors. Flatten $F_{\text{sam}}$ and $F_{\text{vq}}$ into token sequences $Q \in \mathbb{R}^{T \times d_s}$ and $K, V \in \mathbb{R}^{T \times d_s}$ with $T = H'W'$. FAA performs multi-head cross-attention:

$$\text{Attn}(Q, K, V) = \text{softmax}\left(\frac{QK^\top}{\sqrt{d_s}}\right) V. \tag{8}$$

We adopt residual connections, layer normalization(LN), and a feed-forward network(FFN):

$$X_1 = \text{LN}\left(Q + \text{Attn}(Q, K, V)\right), \tag{9}$$
$$X_2 = \text{LN}\left(X_1 + \text{FFN}(X_1)\right). \tag{10}$$

Reshaping $X_2$ back to spatial form yields fused features $F_{\text{fuse}} \in \mathbb{R}^{d_s \times H' \times W'}$.

**Segmentation loss.** We train Stage 2 parameters (segmentation head, SAM adaptation modules if used, and FAA) with a combination of cross-entropy and overlap-aware losses:

$$\mathcal{L}_{\text{seg}} = \alpha \, \mathcal{L}_{\text{CE}}(\hat{p}, y) + \beta \, \mathcal{L}_{\text{Dice}}(\hat{p}, y), \tag{11}$$

where $\alpha, \beta$ control imbalance handling. Importantly, the Stage 1 VQ encoder remains frozen to preserve the learned discrete prior.

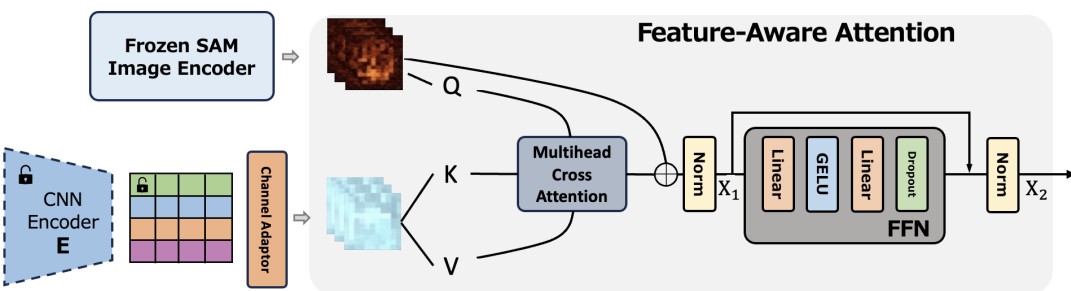

Figure 2: FAA block. SAM embeddings serve as queries and VQ priors as keys/values. Residual + LayerNorm + FFN enable stable training while allowing selective retrieval of texture cues from discrete priors.

---

**Algorithm 1** Training procedure (private-data self-contained).

---

1: **Stage 1: VQ prior learning (train split only)**
2: **for** each minibatch $(x, y)$ from train split **do**
3:     $h \leftarrow E_\phi(x)$; $z_{\text{vq}} \leftarrow \text{VQ}(h)$; $\hat{x} \leftarrow D_\phi(z_{\text{vq}})$
4:     Sample projected features from hard classes; compute $\mathcal{L}_{\text{supcon}}$
5:     Update $\phi$ by minimizing $\mathcal{L}_{\text{VQ}}$ (Eq. 3)
6: **end for**
7: Freeze $E_\phi$ and codebook $\mathcal{Z}$
8: **Stage 2: segmentation training (train split only)**
9: **for** each minibatch $(x, y)$ from train split **do**
10:     $F_{\text{sam}} \leftarrow \text{SAMEnc}(x)$ (with chosen adaptation baseline)
11:     $F_{\text{vq}} \leftarrow A(\text{resize}(\text{VQ}(E_\phi(x))))$
12:     $F_{\text{fuse}} \leftarrow \text{FAA}(F_{\text{sam}}, F_{\text{vq}})$
13:     $\hat{p} \leftarrow \text{SegHead}(F_{\text{fuse}})$
14:     Update trainable parameters by minimizing $\mathcal{L}_{\text{seg}}$
15: **end for**

---

**Training procedure.**

## 4 Experiments

### 4.1 Private Dataset, Splits, and Evaluation Protocol

**Dataset and labels.**    We evaluate on a private multi-class coronary CCTA dataset with expert annotations following  (Huang et al., 2020; Ruan et al., 2025b;a). To preserve double-blind review, patient demographics and scanner/vendor information are omitted. The cohort contains 100 cardiac CCTA volumes. Each volume is annotated slice-wise into $C = 5$ classes: background, coronary lumen, vessel wall, calcified plaque, and non-calcified plaque. All scans are de-identified and used under institutional ethics approval (IRB).

**Annotation procedure.**    Annotations were produced by trained cardiac readers and cross-checked by a second senior reader to improve consistency, with disagreements resolved through review.

**Patient-level split.**    To avoid information leakage across correlated slices from the same patient, we split the cohort at the patient level into 70/10/20 for train/validation/test. Unless stated otherwise, all reported results are computed on the held-out test set, with model selection performed on the validation set.

**Evaluation metrics.** We report class-wise Dice similarity coefficient (Dice) for the four *foreground* classes (lumen, vessel wall, calcified plaque, and non-calcified plaque), and **Macro Avg** as their unweighted mean (background excluded). Dice is the primary metric because it is standard for medical segmentation. For completeness, we also report class-wise IoU and the corresponding **Macro Avg**, computed over the same four foreground classes.

**Scope of evaluation.** We emphasize that this study evaluates one private CCTA cohort with a fixed five-class plaque taxonomy. We do not claim broad cross-domain generalization from the present experiment alone. A practical difficulty is that publicly available coronary CT datasets do not provide a directly matched label space for our target setting, especially for multi-class plaque annotation including non-calcified plaque. For this reason, we position the paper as a controlled study of adaptation behavior in a private-domain setting rather than a broad external-validation paper.

## 4.2 Compared Methods

We compare against both strong medical segmentation backbones and modern SAM-family models. To ensure fair comparison, all methods are trained on the same training split and evaluated under the same preprocessing and postprocessing settings.

**(i) Non-SAM medical segmenters.**

- **nnU-Net** (Isensee et al., 2021): a self-configuring U-Net framework widely used as a robust baseline in medical segmentation.

- **TransUNet** (Chen et al., 2021): a hybrid CNN-Transformer segmentation model that is competitive on many medical benchmarks.

**(ii) SAM-family baselines.** We include the original Segment Anything Model (SAM) and medically adapted variants:

- **SAM** (Kirillov et al., 2023): the foundation vision model for promptable segmentation.

- **Medical SAM** (Ma et al., 2024): a SAM variant adapted to medical images (weights and/or training recipe tuned for medical domains).

- **CAT-SAM** (Xiao et al., 2024): a conditional tuning strategy that provides a strong and stable parameter-efficient SAM adaptation baseline.

**Adaptation settings for SAM-family baselines.** For SAM and Medical SAM, we evaluate two regimes under the same **MC-Head protocol** (a lightweight multi-class head on top of encoder features): **(a) frozen** (train MC-Head only), and **(b) LoRA adaptation** (train low-rank adapters inserted into the image encoder together with MC-Head, keeping the original encoder weights frozen) Hu et al. (2022). We denote them as "(frozen) + MC-Head" and "+ LoRA + MC-Head" in Tables 1–2.

## 4.3 Implementation Details

Our framework consists of two stages: Stage 1 learns a discrete prior, and Stage 2 injects that prior into a SAM-family encoder via a feature-aware cross-attention fusion module (FAA) before multi-class decoding.

**Preprocessing.** All CCTA slices are intensity-normalized and resized/cropped to match the input resolution expected by the SAM-family backbone. We use identical resizing and cropping across all baselines to avoid confounding factors. For training, we apply HU-to-grayscale normalization (HU2Gray), random rotation with probability 0.5, center cropping, rescaling, mask conversion (Gray2Mask), and tensor conversion (ToTensor). The random rotation is applied only on the training split, while the remaining preprocessing steps are applied consistently across splits.

**Stage 1: discrete prior learning (VQ + supervised contrastive).** We train a vector-quantized autoencoder to learn compact discrete tokens from CCTA appearance. To encourage discriminative priors for hard-to-separate classes (especially plaque subtypes), we apply supervised contrastive learning on quantized features, using a class-balanced sampler that emphasizes hard classes. This stage produces a bank of quantized prior features that are later used as *keys/values* in the FAA fusion during Stage 2.

**Stage 2: segmentation training.** We train multi-class segmentation models using AdamW (Loshchilov & Hutter, 2019) with cosine (or step) learning-rate scheduling and early stopping based on validation Dice. For SAM-family models, we keep the backbone variant fixed (ViT-B) across comparisons and only change the adaptation strategy (frozen vs fine-tuning; CAT-SAM vs ours). All models are trained for the same maximum epochs and validated at a fixed frequency.

**Multi-class decoding for SAM-family models.** Since SAM is originally designed for promptable binary masks, we adapt it to multi-class semantic segmentation using two options: (i) a lightweight multi-class head on top of SAM embeddings, and (ii) a multi-class decoder variant that outputs $C$ masks via class-specific tokens. These options are later ablated in Sec. 5.2, and we use the stronger setting for the final comparison.

**FAA fusion (prior injection).** The FAA module is implemented as a query-based cross-attention adapter: SAM/CAT-SAM features act as *queries*, and the Stage 1 discrete prior features serve as *keys/values*, so the segmentation backbone can selectively retrieve texture-specific cues relevant to plaque delineation. We insert FAA at late encoder blocks where semantic features are sufficiently expressive; unless stated otherwise, the insertion location and the number of attention heads are kept fixed across all experiments for a controlled comparison.

### 4.4 Quantitative Results

Tables 1 (Dice; primary) and 2 (IoU; auxiliary) summarize the patient-level test results. Non-SAM segmenters are strong on lumen/wall but lag on fine-grained plaque subtypes (e.g., nnU-Net Dice: non-cal. 32.9 and Macro Avg 62.6 in Table 1). Within the SAM family under the same MC-Head protocol, LoRA yields a modest gain over frozen encoders (SAM Dice Macro Avg 63.2→64.2), and CAT-SAM is a strong baseline (64.3). Injecting the learned discrete prior via FAA improves over CAT-SAM (Dice Macro Avg 66.4 vs. 64.3; non-cal. Dice 41.6 vs. 36.7), with consistent gains in IoU (Macro Avg 52.7 vs. 50.7; non-cal. IoU 26.3 vs. 22.5 in Table 2). Finally, replacing the MC-head with our multi-class decoder further improves performance, especially on non-calcified plaques (Dice: Macro Avg 67.8 and non-cal. Dice 45.2).

The class-wise pattern is also consistent with the task difficulty: the largest gains appear on vessel wall and non-calcified plaque, where boundaries are weaker and the visual evidence is subtler and more variable, while lumen and calcified plaque already provide relatively stronger structural or intensity cues. We therefore interpret the effect of the proposed design as concentrated on the categories that need task-specific texture evidence most, rather than as a uniform gain across all classes.

### 4.5 Qualitative Vision Results

We visualize representative slices in Figure 3. Each row corresponds to one CCTA slice, and columns show the input image, the ground-truth mask, and predictions from (left to right) **Ours**, **CAT-SAM** (multi-class head), **SAM+LoRA** (multi-class head), and **nnU-Net**. Masks use a consistent color legend: **black** = background, **blue** = coronary lumen, **white** = vessel wall, **yellow** = calcified plaque, and **red** = non-calcified plaque.

## 5 Ablation Study

### 5.1 Component Ablations

Table 3 isolates three design choices with a minimal factorial design: (i) the multi-class output design (multi-class head vs. multi-class decoder), (ii) the fusion mechanism (FAA cross-attention vs. additive fusion),

Table 1: Dice comparison on the private CCTA dataset (patient-level split). Macro Avg is the unweighted mean over the four foreground classes (lumen, wall, calcified plaque, and non-calcified plaque).

| Method | Lumen | Wall | Cal. Plaque | Non-cal. Plaque | Macro Avg |
|---|---|---|---|---|---|
| **Non-SAM medical segmenters** | | | | | |
| nnU-Net | 86.4 | 53.9 | 77.3 | 32.9 | 62.6 |
| TransUNet | 87.2 | 53.3 | 77.5 | 31.5 | 62.3 |
| **SAM-family baselines (MC-Head protocol)** | | | | | |
| SAM (frozen) + MC-Head | 87.3 | 54.3 | 77.5 | 33.9 | 63.2 |
| SAM + LoRA + MC-Head | 87.5 | 55.5 | 78.4 | 35.5 | 64.2 |
| Medical SAM (frozen) + MC-Head | 87.9 | 53.4 | 76.3 | 34.2 | 62.9 |
| Medical SAM + LoRA + MC-Head | 88.5 | 53.3 | 77.2 | 35.0 | 63.5 |
| CAT-SAM + MC-Head | **89.1** | 54.3 | 77.1 | 36.7 | 64.3 |
| **Ours (CAT-SAM backbone + injections)** | | | | | |
| Ours: VQ prior + additive fusion (no attention) | 88.5 | 55.1 | 79.0 | 39.4 | 65.5 |
| Ours: VQ prior + FAA (cross-attention fusion) | 88.7 | 56.3 | **79.2** | 41.6 | 66.4 |
| Ours: VQ prior + FAA + multi-class decoder (final) | 88.5 | **59.1** | 78.5 | **45.2** | **67.8** |

Table 2: IoU comparison on the private CCTA dataset (patient-level split). Macro Avg is the unweighted mean over the four foreground classes (lumen, wall, calcified plaque, and non-calcified plaque).

| Method | Lumen | Wall | Cal. Plaque | Non-cal. Plaque | Macro Avg |
|---|---|---|---|---|---|
| **Non-SAM medical segmenters** | | | | | |
| nnU-Net | 76.1 | 36.9 | 63.0 | 19.7 | 48.9 |
| TransUNet | 77.3 | 36.3 | 63.3 | 18.7 | 48.9 |
| **SAM-family baselines (MC-Head protocol)** | | | | | |
| SAM (frozen) + MC-Head | 77.5 | 37.3 | 63.3 | 20.4 | 49.6 |
| SAM + LoRA + MC-Head | 77.8 | 38.4 | 64.5 | 21.6 | 50.6 |
| Medical SAM (frozen) + MC-Head | 78.4 | 36.4 | 61.7 | 20.6 | 49.3 |
| Medical SAM + LoRA + MC-Head | 79.4 | 36.3 | 62.9 | 21.2 | 49.9 |
| CAT-SAM + MC-Head | **80.3** | 37.3 | 62.7 | 22.5 | 50.7 |
| **Ours (CAT-SAM backbone + injections)** | | | | | |
| Ours: VQ prior + additive fusion (no attention) | 79.4 | 38.0 | 65.3 | 24.5 | 51.8 |
| Ours: VQ prior + FAA (cross-attention fusion) | 79.7 | 39.2 | **65.6** | 26.3 | 52.7 |
| Ours: VQ prior + FAA + multi-class decoder (final) | 79.4 | **41.9** | 64.6 | **29.2** | **53.8** |

and (iii) supervised contrastive structuring of the discrete prior (w/o SupCon, SupCon 5-class, and SupCon 3-class on hard classes). All ablations use the same backbone and training schedule as the final method.

**Discussion.** Table 3 indicates that the gain is concentrated in the hard foreground categories, particularly non-calcified plaque. Comparing **Full** to **A1** isolates the effect of multi-class decoding under the same (FAA + SupCon 3-class) setting: the multi-class decoder improves the Dice of *vessel wall* and *non-calcified plaque*, while the lightweight head is slightly higher on *lumen* and *calcified plaque*. This suggests that the decoder mainly helps where wall/plaque ambiguity is strongest, whereas the lighter head is already sufficient for easier classes. Comparing **Full** to **A2** keeps the decoder fixed and replaces FAA with additive fusion: additive fusion remains competitive on lumen/wall/calcified plaque, but it yields a lower non-calcified plaque Dice, which reduces the overall Macro Avg. This is consistent with the intended role of FAA: additive fusion injects prior features everywhere, while query-based fusion allows the backbone to retrieve prior cues more

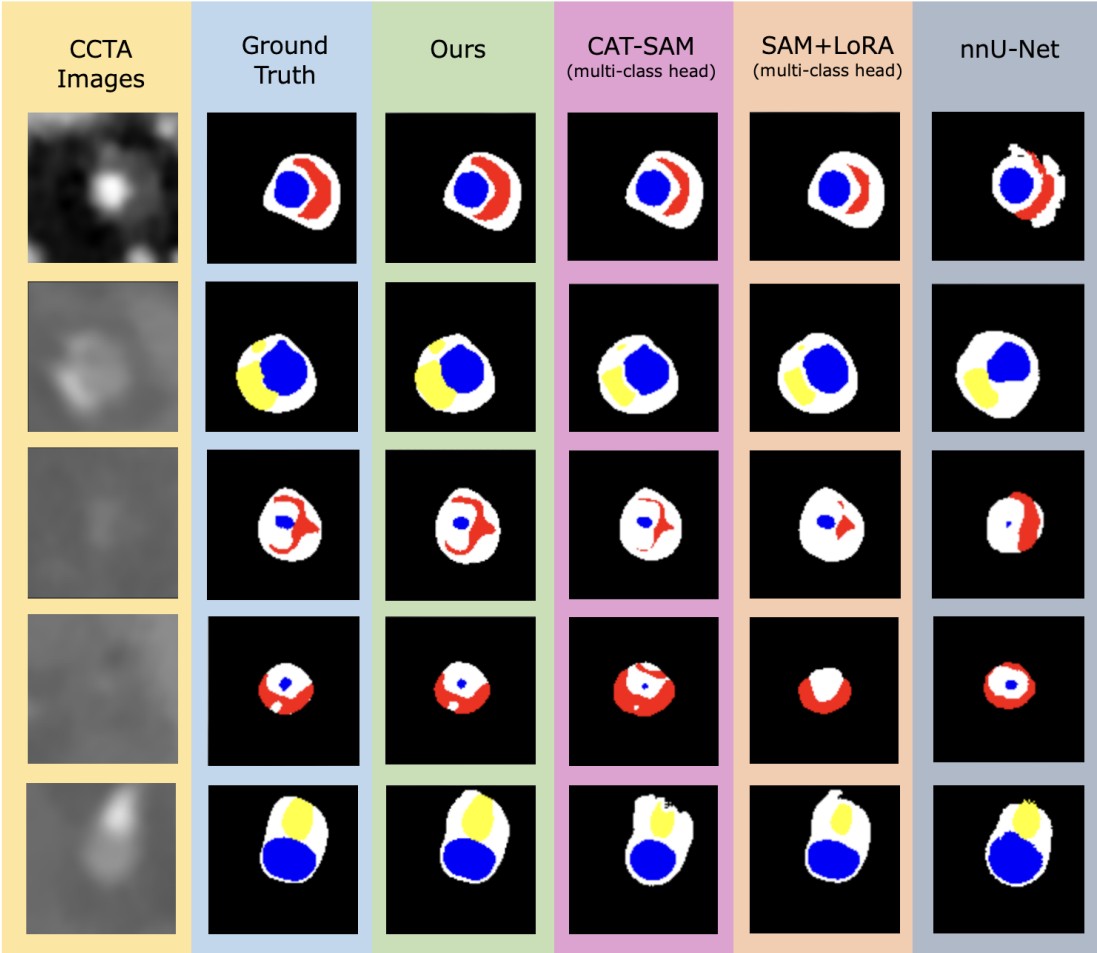

Figure 3: Qualitative comparison across methods. Each row shows one example slice: input CCTA, ground truth, and predictions from Ours, CAT-SAM (multi-class head), SAM+LoRA (multi-class head), and nnU-Net. The masks use the following color legend: Blue denotes lumen, white denotes vessel wall, yellow denotes calcified plaque, and red denotes non-calcified plaque.

selectively where extra texture evidence is needed. Finally, block **B** fixes (FAA + multi-class head) and varies the supervised contrastive objective: removing SupCon (**B1**) decreases all foreground scores, and focusing SupCon on the three hard categories (**B3**) achieves higher scores than applying it to all five categories (**B2**), matching the 3-class vs. 5-class comparison in Table 3. This suggests that hard-class-focused SupCon is more useful than distributing the contrastive objective over easier background/lumen patterns. Taken together, these comparisons suggest that the final gain is distributed across three components rather than coming from a single change alone. In particular, SupCon improves the usefulness of the learned prior, FAA helps convert that prior into selective feature retrieval, and the multi-class decoder provides a better output parameterization for the hardest categories.

## 5.2 Decoder and Multi-class Output Ablations

Comparing **A1** to **Full** in Table 3, replacing the multi-class decoder with a lightweight head lowers the overall Macro Avg and reduces the Dice on vessel wall and non-calcified plaque under the same (FAA + SupCon 3-class) setting. Since SAM is designed for promptable binary masks, we implement two deterministic options for $C$-way semantic outputs: (i) a lightweight multi-class head on top of encoder embeddings, and (ii) a multi-class decoder that outputs $C$ masks via class-specific tokens. We use the multi-class decoder in the final

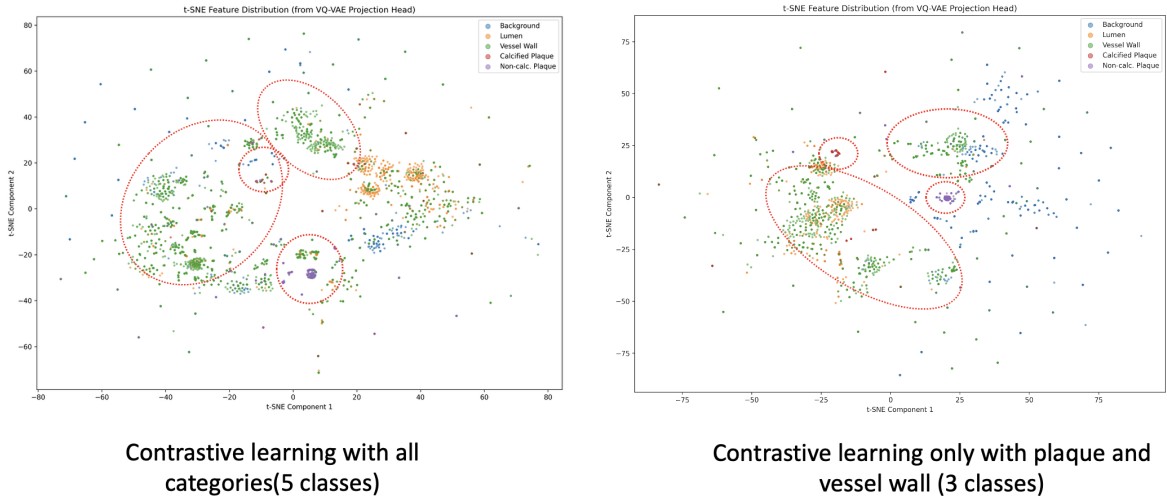

Figure 4: t-SNE visualization of quantized prior features under two supervised contrastive settings: (left) SupCon over all categories (5 classes), and (right) SupCon only over plaque and vessel wall (3 classes).

Table 3: Ablations with a minimal but informative factorial design. We anchor on the full model and isolate the effects of (i) multi-class decoding, (ii) fusion mechanism (FAA vs. additive), and (iii) contrastive structuring of the discrete prior, including a 3-class vs. 5-class SupCon comparison. Macro Avg is the unweighted mean over the four foreground classes.

| Variant | Lumen | Wall | Cal. Plaque | Non-cal. Plaque | Macro Avg |
|---|---|---|---|---|---|
| **Full: VQ prior + SupCon (3-class) + FAA + multi-class decoder** | 88.5 | 59.1 | 78.5 | 45.2 | 67.8 |
| **(A) Decoding & Fusion (with fixed SupCon 3-class)** | | | | | |
| A1: FAA + multi-class *head* (replace decoder) | 88.7 | 56.3 | 79.2 | 41.6 | 66.4 |
| A2: Additive fusion + multi-class decoder (remove FAA) | 88.4 | 59.4 | 78.8 | 41.8 | 67.1 |
| A3: Additive fusion + multi-class *head* (remove FAA, replace decoder) | 88.8 | 57.3 | 78.9 | 38.9 | 65.9 |
| **(B) Contrastive structuring of the prior (with fixed FAA + multi-class head )** | | | | | |
| B1: w/o SupCon ($\lambda_{con} = 0$) | 87.6 | 55.8 | 78.1 | 34.9 | 64.1 |
| B2: SupCon (5-class) | 88.6 | 56.1 | 78.5 | 38.3 | 65.4 |
| B3: SupCon (3-class) | 88.7 | 56.3 | 79.2 | 41.6 | 66.4 |

model because, under the same FAA + SupCon 3-class setting, it improves the hardest categories without changing any other components, suggesting a more suitable class-specific parameterization for ambiguous vessel-wall and non-calcified-plaque predictions.

### 5.3 Latent Space Analysis

We visualize the Stage-1 quantized prior features using t-SNE (van der Maaten & Hinton, 2008) in Figure 4. The left panel applies supervised contrastive learning over all categories ($C$=5), while the right panel applies it only to vessel wall and plaque categories ($C$=3), matching the SupCon settings compared in Table 3. Points are colored by semantic class.

### 5.4 Limitation and Scope

This study has several limitations. First, the evaluation is restricted to a single private CCTA cohort, so the current evidence supports the method only within this setting rather than broad cross-domain generalization. Second, because publicly available coronary CT datasets do not provide a directly matched multi-class plaque

taxonomy, especially for non-calcified plaque, we are unable to include a clean external validation benchmark without changing the task definition itself. Third, while the ablations support the usefulness of the discrete prior, FAA, and multi-class decoder, the present study does not claim to fully separate all sources of gain beyond this controlled setting.

## 6    Conclusion

We introduced a performance-driven framework for multi-class coronary plaque segmentation in CCTA that learns class-structured discrete latent priors and injects them into a SAM-based encoder via feature-aware attention fusion. By organizing VQ latents with supervised contrastive learning focused on hard classes and enabling selective retrieval through cross-attention, the proposed approach reduces wall/plaque ambiguity and improves plaque segmentation metrics. Extensive comparisons against strong non-SAM baselines and multiple SAM adaptation strategies demonstrate that the discrete prior, FAA fusion, and multi-class decoder are useful contributors to the final performance in this private CCTA setting, with the largest gains appearing on vessel wall and non-calcified plaque. More broadly, we position the present work as a targeted private-domain adaptation study rather than a claim of broad cross-domain generalization.

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
