# Appendix: Contrastive VQ Priors for Multi-Class Plaque Segmentation via SAM Adaptation

**Yizhe Ruan**[1,2], **Yusuke Kurose**[1,2], **Junichi Iho**[3],
**Yoji Tokunaga**[3], **Makoto Horie**[3], **Yusaku Hayashi**[3], **Keisuke Nishizawa**[3],
**Yasushi Koyama**[3,2], **Tatsuya Harada**[1,2]
[1]**The University of Tokyo**
[2]**RIKEN Center for Advanced Intelligence Project**
[3]**Sakurabashi Watanabe Advanced Healthcare Hospital**
ruanyizhe@mi.t.u-tokyo.ac.jp

**Reviewed on OpenReview:** `https://openreview.net/forum?id=5P7HfuejgL`

## 1 Appendix

### 1.1 Additional Qualitative Results

Figure 1 reports qualitative comparisons across methods. Each row corresponds to one representative CCTA slice. From left to right, we show the input image, the ground-truth mask, and predictions from **Ours**, **CAT-SAM** (multi-class head), **SAM+LoRA** (multi-class head), and **nnU-Net**. We use a consistent color scheme throughout the paper: **black** indicates background, **white** denotes vessel wall, **blue** denotes lumen, **yellow** denotes calcified plaque, and **red** denotes non-calcified plaque. The last three rows (highlighted in orange) correspond to visually challenging cases, where the predictions are generally less reliable yet remain comparable across methods.

### 1.2 Training and Reproducibility Details

**Hardware and software.** Experiments are run on a single machine with 8 NVIDIA Tesla V100-SXM2 GPUs (32 GB each). The NVIDIA driver version is 535.247.01 and the CUDA version is 12.2. All models are implemented in PyTorch and trained with distributed data parallel (DDP).

**Data split and selection.** We use patient-level train/validation/test splits. All model selection and hyperparameter choices are made using the validation set only. The final checkpoint is chosen by the best validation score and evaluated once on the held-out test set.

**Preprocessing and augmentation.** CCTA slices are normalized to a fixed intensity range and resized/cropped to the network input resolution. During training, we apply mild geometric augmentation (random rotation with probability 0.5). All methods use the same preprocessing and augmentation pipeline unless stated otherwise.

**Batch size, epochs, and early stopping.** We use a per-GPU batch size of 8 (effective batch size 64 with 8 GPUs). Training runs for up to 100 epochs with early stopping based on validation macro Dice (background excluded). Validation is performed every 5 epochs. Early stopping uses patience $P = 6$ validations (i.e., training stops after 6 consecutive validations without improvement).

### 1.3 Stage-1: Contrastive VQ Prior Learning

**Model.** Stage-1 trains a 2D vector-quantized model (VQ) with an encoder–codebook–decoder structure. The codebook size is $K = 1024$ and the latent channel dimension is $d = 32$. A small projection head maps latent features to a contrastive embedding space.

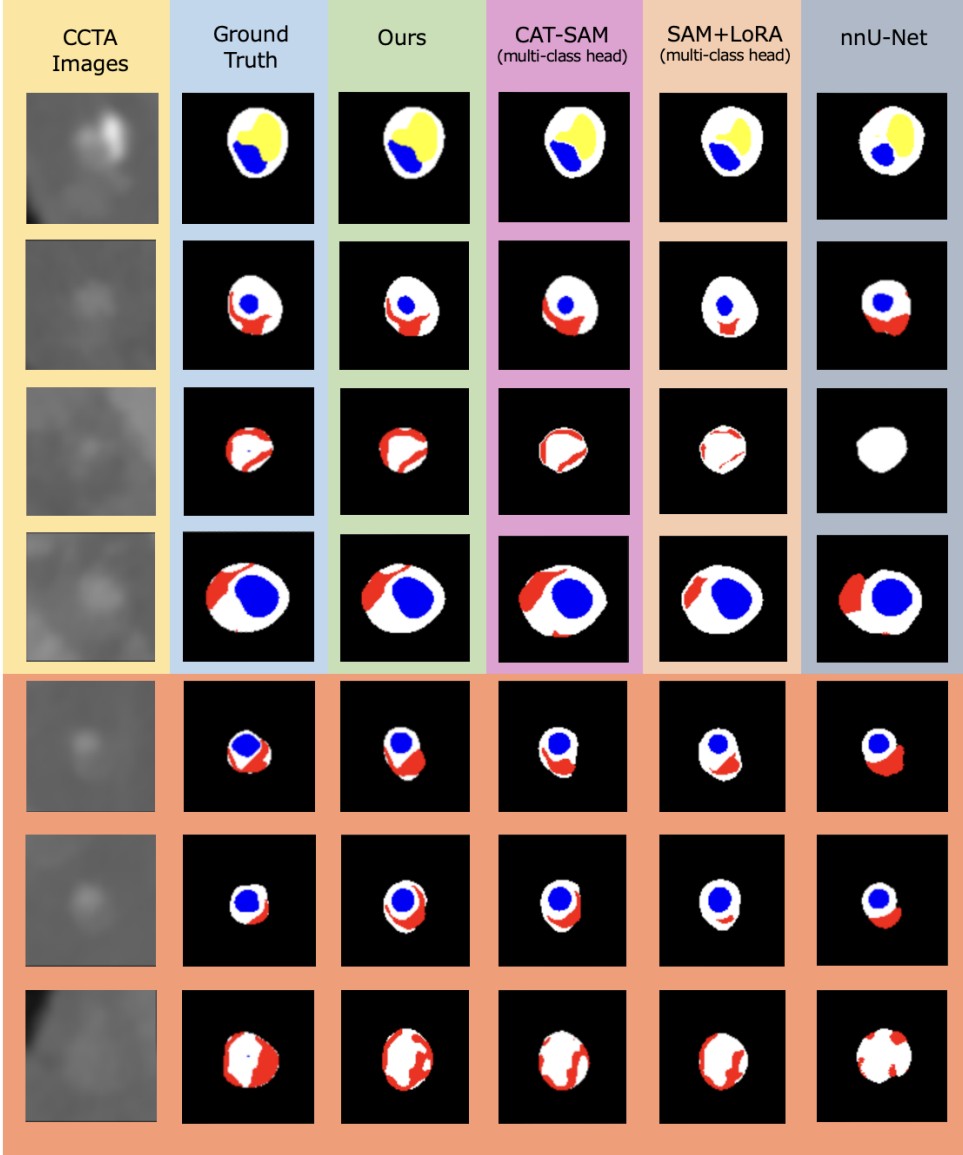

Figure 1: Qualitative comparison across methods. Each row shows one example slice: input CCTA, ground truth, and predictions from Ours, CAT-SAM (multi-class head), SAM+LoRA (multi-class head), and nnU-Net. The last three rows (highlighted in orange) correspond to visually challenging but comparable cases.

**Objective.** Stage-1 optimizes reconstruction, perceptual similarity, VQ commitment, and supervised contrastive learning:

$$\mathcal{L}_{S1} = \lambda_1 \mathcal{L}_1 + \lambda_p \mathcal{L}_p + \lambda_c \mathcal{L}_{commit} + \lambda_{con} \mathcal{L}_{con}.$$

We use $\lambda_1 = 1.0$, $\lambda_p = 1.0$, $\lambda_c = 0.25$, $\lambda_{con} = 5.0$, and temperature $\tau = 0.1$.

**Hard-class sampling for supervised contrastive learning.** To focus on clinically challenging regions, the supervised contrastive term samples features from a subset of hard classes (e.g., vessel wall and plaque subtypes). Given a projected feature map and a label map aligned to the latent resolution, we sample up to $N = 2048$ feature vectors per selected class per batch (with replacement when insufficient pixels exist), forming balanced positive/negative pairs.

**Optimization.** Stage-1 uses Adam with learning rate $2 \times 10^{-4}$ and betas $(0.5, 0.999)$. We train for up to 100 epochs with the early-stopping protocol in Appendix 1.2. After Stage-1, the VQ encoder and codebook are frozen for Stage-2 to preserve the learned discrete representation.

### 1.4 Stage-2: Prior Injection with FAA and Multi-class Decoding

**Prior alignment.** The frozen VQ encoder produces a quantized latent map. We resize it to the backbone embedding resolution and apply a $1 \times 1$ adapter to match channel dimensions before fusion.

**FAA block.** FAA performs multi-head cross-attention where the segmentation backbone features provide queries and the VQ prior provides keys/values. We use 4 attention heads and dropout 0.1. The block follows a residual + normalization + feed-forward structure:

$$F_{\text{fuse}} = F_{\text{sam}} + \text{MHCA}(\text{LN}(F_{\text{sam}}), \text{LN}(F_{\text{vq}})), \quad F_{\text{out}} = F_{\text{fuse}} + \text{FFN}(\text{LN}(F_{\text{fuse}})).$$

**Multi-class decoding.** We use two decoding options for multi-class segmentation: (i) a lightweight multi-class head on top of fused features, and (ii) a class-token decoder that predicts $C$ masks using class-specific tokens, producing logits $\hat{Y} \in R^{C \times H \times W}$.

**Stage-2 loss and optimization.** Stage-2 combines multi-class cross-entropy and an overlap-aware term (Dice/Tversky family):

$$\mathcal{L}_{\text{seg}} = \lambda_{\text{ce}} \mathcal{L}_{\text{CE}} + \lambda_{\text{dice}} \mathcal{L}_{\text{dice}},$$

with $\lambda_{\text{ce}} = 1.0$ and $\lambda_{\text{dice}} = 1.0$. We optimize with AdamW (learning rate $1 \times 10^{-4}$, weight decay $1 \times 10^{-4}$) using the same epoch budget and early-stopping policy as Appendix 1.2.