# OpenReview forum: "Contrastive VQ Priors for Multi-Class Plaque Segmentation via SAM Adaptation"
_TMLR — Accepted by TMLR_

### Review · Reviewer_H6Ps · 2026-03-09

**Summary Of Contributions:**

The paper proposes a general recipe for adapting foundation models to medical imaging:

- learn task-specific discrete priors
- structure them with contrastive learning
- inject them into SAM via cross-attention

The paper improves coronary plaque segmentation by learning texture-aware discrete priors with VQ-VAE + contrastive learning and injecting them into SAM through cross-attention, enabling better separation of subtle plaque types.

**Audience:**

Yes

**Audience Explanation:**

The paper addresses an important problem in medical image segmentation, namely improving plaque subtype delineation in coronary CT angiography under limited annotation settings. The proposed framework combines foundation model adaptation with domain-specific representation learning, which may be of interest to researchers working on medical imaging, segmentation foundation models, and domain adaptation for specialized tasks. In particular, the idea of injecting learned task-specific priors into a pretrained segmentation backbone could be relevant to practitioners attempting to transfer large vision models to private or small-scale medical datasets. While the methodological novelty is moderate, the empirical findings and the design pattern for combining discrete priors with foundation models may still provide useful insights for the TMLR audience, especially those working at the intersection of representation learning and applied medical AI.

**Claims And Evidence:**

No

**Claims Explanation:**

From my understanding, it is more of a solid systems/engineering paper than a fundamentally new algorithmic paper.

Most components of the framework are built upon established techniques, including VQ-VAE for discrete latent representation learning, supervised contrastive learning for structuring feature spaces, and cross-attention for feature fusion. The primary contribution, therefore, lies in the integration of these existing components into a two-stage pipeline for adapting SAM to a specialized medical segmentation task. While this design is sensible and empirically effective, it represents more of an application-driven architectural composition rather than the introduction of a fundamentally new learning principle or model class. In particular, the improvement may partially stem from the additional domain-specific representation learning provided by the VQ prior, rather than from a fundamentally new mechanism for foundation-model adaptation. Furthermore, the evaluation is conducted on a relatively small private dataset, which makes it difficult to assess whether the proposed design pattern generalizes broadly beyond this specific task setting. Overall, the work can be viewed as a carefully engineered adaptation strategy that combines known techniques in a coherent way, rather than a conceptually novel methodological advance.

**Requested Changes:**

- Clarify the source of performance gains.

The current method combines several components (VQ prior learning, supervised contrastive structuring, FAA cross-attention, and multi-class decoding). While the ablation study partially isolates their contributions, it remains unclear how much improvement comes specifically from the proposed prior-injection mechanism versus simply introducing additional domain-specific representation learning. The authors should provide clearer analysis or additional ablations to demonstrate that the gains are not primarily due to increased model capacity or additional pretraining on the target dataset.

- Strengthen evaluation and generalization evidence.

The experiments are conducted on a single private dataset consisting of 100 CCTA volumes. While this is understandable for clinical data, the limited scope makes it difficult to assess the broader applicability of the proposed method. If possible, evaluation on an additional dataset, cross-institution split, or other external validation would substantially strengthen the claim that the method generalizes beyond the specific dataset used.

- Clarify methodological novelty and positioning.

The method combines several established techniques (VQ-VAE, supervised contrastive learning, and cross-attention fusion). The paper would benefit from a clearer articulation of what aspect of the framework constitutes the primary methodological contribution. Explicitly positioning the work relative to existing SAM adaptation strategies and prior-based segmentation approaches would help readers better understand the novelty and scope of the contribution.

---

> ### Author Response · Authors · 2026-03-26
> **Rebuttal**
>
> We thank the reviewer for the thoughtful comments and for clearly identifying the main positioning questions of the paper. We agree that the work is best presented as a carefully engineered adaptation study for a hard private-domain medical segmentation task, rather than as a fundamentally new algorithmic family.
>
> **(1) Source of performance gains.**
> We agree that the gains should not be attributed to a single component without care. To address this, we strengthened the ablation discussion and made the decomposition more explicit. Our current interpretation is that the final improvement is distributed across three components rather than coming from one change alone:
>
> * supervised contrastive learning improves the usefulness of the learned prior,
> * FAA converts that prior into selective feature retrieval, and
> * the multi-class decoder provides a better output parameterization for the hardest categories.
>
> We also softened the strongest causal wording in the paper accordingly. Rather than claiming that the gains are “not merely due to additional training,” we now state that the ablations **support the usefulness** of these components within this setting.
>
> **(2) Generalization evidence.**
> We agree that broader external validation would strengthen the paper. However, this task has a practical constraint: for our target setting, publicly available coronary CT datasets do not provide a directly matched **multi-class plaque taxonomy**, especially for non-calcified plaque. Because of this, a clean external validation benchmark is not available without materially changing the task definition itself. We therefore revised the manuscript to state this limitation explicitly and to position the paper as a **controlled private-domain study** rather than a broad cross-domain generalization claim.
>
> **(3) Methodological novelty and positioning.**
> We agree with the reviewer that the main contribution is not a new standalone attention primitive or a new VQ formulation in isolation. We revised the wording throughout the paper to reflect a narrower and more accurate framing: the contribution is a **specific integration** of class-structured discrete priors, query-based prior injection, and a multi-class output design for private CCTA plaque segmentation. We believe this is the fairest description of the present evidence.
>
> We appreciate the reviewer’s comments, which helped us substantially improve the positioning and claim calibration of the manuscript.

---

### Review · Reviewer_qA9N · 2026-03-14

**Summary Of Contributions:**

This paper proposes a method for a task in medical imaging --- plaque segmentation in coronary CT. The main motivating observation of this paper is that tradition segmentation methods like UNet may suffer from sparse data and do not perform well. The proposed method resorts to SAM models --- foundation models in image segmentation, aiming to transfer their robust segmentation ability into this specific domain. The proposed method involves training VQ embedding codes for CCTA images with autoencoding and contrastive objectives, and fuse the learned codes with SAM features with cross-attention. Experiments on a private CCTA dataset shows the effectiveness of the proposed method, with good accuracy and extensive ablations.

**Audience:**

Yes

**Audience Explanation:**

This paper focuses on a specific application of image segmentation and I think those who work on this specific area should be interested in this paper. Moreover, those who work on a broader topic of medical image segmentation may also find interest in this paper as most challenges are shared.

**Broader Impact Concerns:**

No concerns since the authors have already discussed responsible data usage.

**Claims And Evidence:**

Yes

**Claims Explanation:**

I am not an expert in this area of medical image segmentation, but from what the paper says, the paper seems sound (in terms of adapting a foundation model to a specific domain with limited data).

The main claim of this paper, from my understanding, is that naive fine-tuning is not a good solution to this problem because it may risk overfitting and compromise the general ability. The authors made comprehensive comparisons with methods along this line (SAM-family baselines).

Other techniques proposed in this paper are conceptually simple but makes sense, and backed up with ablation studies (Table 3) which is sufficient for an application-oriented paper.

**Requested Changes:**

No in my opinion, since I am not an expert in this area.

---

> ### Author Response · Authors · 2026-03-26
> **Rebuttal**
>
> We thank the reviewer for the positive assessment and for recognizing the paper as a sound application-oriented study with comprehensive comparisons and ablations.
>
> Following the broader feedback from the review process, we have further revised the manuscript in three directions that we believe also strengthen the presentation for this reviewer:
>
> 1. We narrowed the wording of the paper so that it is no longer framed as a broad transfer-learning claim, but more explicitly as a **targeted study on private multi-class CCTA plaque segmentation**.
> 2. We improved the alignment between the method text and the figures, especially around the notation and the FAA block.
> 3. We added clearer discussion of the scope and limitations of the evaluation, including why a directly matched public external-validation benchmark is not available for this task setting.
>
> We appreciate the reviewer’s encouraging feedback and believe the revised version now presents the contribution with a more precise scope and clearer exposition.

---

### Review · Reviewer_eE3p · 2026-03-16

**Summary Of Contributions:**

The paper proposes a two-stage transfer learning framework for adapting the Segment Anything Model (SAM) to new domains. A Vector Quantized Autoencoder (VQ-AE) is trained to encode domain-specific representations, whose latent codes are then injected into SAM via cross-attention. The two-stage design decouples codebook learning from segmentation adaptation, enabling modular and stable training without modifying the pretrained backbone.

Strengths:
* The core mechanism of bridging a VQ-AE to SAM through cross attention builds on well-established components. This makes design choices easy to follow.
* Thorough experimental section with comprehensive ablations and comparison to relevant methods.

Weaknesses:
* Despite a simple idea, section 3 (methods) and figure 1 and 2 are suffer from several minor but cumulative issues that hinder readability. Notation is introduced without sufficient motivation.
*  The paper frames its contribution as a general transfer learning framework for SAM, yet all experiments are confined to a single narrow domain. The paper would greatly benefit from experiments or comments on the expected generalization performance.

**Additional Comments:**

The notation in the VQ-AE description is somewhat confusing. In a standard autoencoder, an encoder E typically maps directly to a latent vector z. Here, the authors instead introduce an intermediate variable h as the output of E, which then passes through VQ to produce z_q. Since h is never used independently in any subsequent derivation, this separation seems unnecessary. Furthermore, using VQ to denote only the quantization step may cause confusion with the VQ-AE model as a whole. The authors should clarify whether this notation follows established convention in the VQ literature, and if not, consider simplifying it. Additionally, the subscript q in z_q risks being misread as referring to the query in the attention mechanism, and an alternative symbol should be considered.

The ablation study shows that the multi-class decoder has a surprisingly large impact on performance. The authors do not discuss this result, and it is not immediately obvious why this component contributes so substantially. The authors should provide some intuition or analysis for why the multi-class decoder drives such a large portion of the gain, as this would significantly aid the reader's understanding of where the method's strength actually lies.

**Audience:**

Yes

**Audience Explanation:**

I am in doubt, this reads to be very niche to the reviewer especially without any consideration of how well the proposed method would generalize.
However, as accordingly to the evaluation guideline, the reviewer will answer yes when in doubt.

**Claims And Evidence:**

Yes

**Claims Explanation:**

The authors provide an extensive ablation experiment detailing their justification for each architecture choice.

**Requested Changes:**

Improve alignment between Section 3, Figure 1, and Figure 2.
The figures and the methods text are poorly connected, making it difficult to follow the description of the method. None of the notation introduced in Section 3 is reflected in the figures, for example, X1 and X2 do not appear in Figure 1 or 2, and the modifications to the multi-class decoder are not visualized at all. Figure 2 goes some way toward illustrating the cross-attention mechanism, but the alignment of inputs makes the roles of Q, K, and V ambiguous, even if they can be inferred. The authors should revise the figures to directly mirror the notation used in the text.

The authors motivate their framework partly by arguing that directly finetuning SAM is problematic in low-data regimes. However, a VQ-AE is itself a data-hungry model, and this tension is never addressed. The paper should provide a clear justification for why a VQ-AE is nevertheless appropriate for this setting, whether through theoretical argument, empirical evidence, or a discussion of the relative data requirements of each component.

The reported improvements are largely concentrated in wall and non-calcified plaque segmentation, while calcified plaque and lumen show minimal gains and in some ablation configurations even regress. The authors do not discuss this disparity at all. What is it about wall and non-calcified plaque that makes them particularly amenable to the proposed approach? And conversely, why does the method seemingly fail to benefit, or occasionally hurt, performance on calcified plaque and lumen? A discussion of the underlying reasons, whether related to visual characteristics, class frequency, boundary ambiguity, or something else, is needed to understand the actual scope and limitations of the contribution.

Additionally, Figure 3 uses color coding to distinguish structures, but no legend or explanation is provided. The authors should clarify the color scheme and ideally include annotated examples that illustrate the visual differences between these structure types, which would help ground the discussion of why gains are unevenly distributed.

---

> ### Author Response · Authors · 2026-03-26
> **Rebuttal**
>
> We thank the reviewer for the detailed and constructive comments. We have revised the manuscript to improve the alignment between the method description, notation, and figures, and to narrow the scope of our claims.
>
> **(1) Alignment between Section 3, Figure 1, and Figure 2.**
> We revised the figures so that they now more directly mirror the notation used in Section 3. In particular, Figure 2 now explicitly visualizes the roles of **Q, K, and V**, includes the intermediate states **(X_1)** and **(X_2)**, and shows the **multi-class decoder** instead of leaving it implicit. We also simplified the notation around the quantized latent to reduce ambiguity with the query notation in attention.
>
> **(2) Scope of the paper.**
> We agree that the earlier wording could be read too broadly. Our intention is not to claim a universally general SAM transfer framework across domains. We therefore revised the abstract, introduction, method, and conclusion to position the paper more narrowly as a **targeted private-domain adaptation study for one multi-class CCTA plaque-segmentation setting**.
>
> **(3) Why a VQ prior is still reasonable in this low-data setting.**
> We agree that a VQ-AE is not “free” from data requirements, and we do not claim that it is universally preferable in all low-data regimes. Our motivation is narrower: Stage 1 learns a compact domain prior from the same training split only, using dense reconstruction and a focused contrastive signal on the hard classes, while keeping this prior learning decoupled from Stage 2 segmentation adaptation. In our setting, this serves as a way to inject domain-specific plaque cues into a strong pretrained encoder without directly relying on broader end-to-end finetuning of the foundation backbone.
>
> **(4) Uneven gains across classes.**
> We agree that the gains are not uniform across all categories, and we now state this more explicitly. The strongest improvements are concentrated on **vessel wall** and **non-calcified plaque**, while lumen and calcified plaque show more modest changes. We interpret this pattern as consistent with the intended role of the prior and FAA module: they are most helpful where texture ambiguity and wall/plaque confusion are hardest.
>
> **(5) Multi-class decoder contribution.**
> We also strengthened the discussion around the decoder ablation. Our interpretation is not that the decoder is a generally novel component by itself, but that, in this task, a class-specific multi-class decoder is a better output parameterization for the hardest categories than a lightweight shared head, which explains why part of the gain appears there.
>
> **(6) Figure 3 legend and visual explanation.**
> We added explicit legend information for the qualitative figure and clarified the color scheme in the text, to make the per-class visual comparison easier to follow.
>
> We appreciate the reviewer’s comments, which helped us improve both the readability and the claim calibration of the paper.

---

### Decision · Action_Editor_MZTj · 2026-04-15

**Recommendation:** Accept with minor revision

**Additional Comments:**

In my view, the requested minor revision is mainly about polishing exposition and claim calibration, rather than requiring additional experiments. Most of the substantive reviewer concerns appear to have been addressed in the revised manuscript already.

For the final version, I would ask the authors to: (1) consistently maintain the narrowed framing of the paper as a targeted study on private multi-class CCTA plaque segmentation, rather than a broad claim about SAM adaptation or cross-domain robustness; (2) further strengthen the discussion of the class-wise results and ablations, especially by adding more intuition for why the gains are concentrated on vessel wall and non-calcified plaque, why lumen and calcified plaque benefit less, and why the multi-class decoder contributes substantially; (3) keep the novelty/contribution statement precise, i.e., frame the gains as arising from the combination of learned priors, attention-based fusion, and decoder design, without overstating methodological novelty; and (4) preserve explicit discussion of the evaluation scope and limitations, including the absence of a directly matched external benchmark.

The figure/notation alignment issues and the qualitative-figure legend seem to have been largely addressed already in the revision, so I would not require new experiments for the minor revision.

**Audience:**

Yes

**Audience Explanation:**

The paper is specialized, but the problem is clinically meaningful and the technical approach should be of interest to readers working on medical image segmentation, adaptation of pretrained segmentation models, and dense prediction under limited private data. This is not a broad-interest paper, but it is clearly relevant to a real segment of the TMLR audience.

**Claims And Evidence:**

Yes

**Claims Explanation:**

In its revised form, the paper makes a much narrower claim than the original submission, and within that narrower scope the empirical support is adequate. The experiments are reasonably thorough for the setting: the paper compares against strong medical segmentation baselines and several SAM-based adaptations, and the ablations are detailed enough to support the main design choices. I do not think the evidence supports a broad claim about SAM adaptation in general or about robustness across domains. But I do think it supports the more limited claim that this particular combination of learned priors, attention-based fusion, and multi-class decoding is effective for the private multi-class CCTA setting studied here.